# Insights into Online microRNA Bioinformatics Tools

**DOI:** 10.3390/ncrna9020018

**Published:** 2023-03-06

**Authors:** Diana Luna Buitrago, Ruth C. Lovering, Andrea Caporali

**Affiliations:** 1BHF Centre for Cardiovascular Science, The Queen’s Medical Research Institute, University of Edinburgh, Edinburgh EH164TJ, UK; 2Functional Gene Annotation, Institute of Cardiovascular Science, University College London, London WC1E 6BT, UK

**Keywords:** miRNAs, bioinformatics, database, web-based tools

## Abstract

MicroRNAs (miRNAs) are members of the small non-coding RNA family regulating gene expression at the post-transcriptional level. MiRNAs have been found to have critical roles in various biological and pathological processes. Research in this field has significantly progressed, with increased recognition of the importance of miRNA regulation. As a result of the vast data and information available regarding miRNAs, numerous online tools have emerged to address various biological questions related to their function and influence across essential cellular processes. This review includes a brief introduction to available resources for an investigation covering aspects such as miRNA sequences, target prediction/validation, miRNAs associated with disease, pathway analysis and genetic variants within miRNAs.

## 1. Introduction

Much of our understanding of miRNAs and their biological roles derives from experimental findings, which have contributed to advancements in the field of miRNA research. However, it is acknowledged that, despite the extensive studies carried out over recent decades, there is still a great need to understand more regarding the total number of existing miRNAs and the full impact they have within regulatory pathways. In recent years, there has been an expansion of the number of bioinformatic tools assisting in the massive demand in the field [1]. Currently, some tools aid researchers in seeking information regarding various areas, including miRNA identification, the prediction of miRNA targets and known associations with disease [2]. Online web-based tools, in particular, provide additional benefits to researchers and groups less experienced with programming. They are more associated with being user-friendly; therefore, they can be accessed by a more extensive range of audiences. In this brief review, a selection of such tools are categorized according to their primary purpose and will be introduced to highlight the diverse roles that miRNAs play. Tools that display a fully functioning web interface with available documentation and publications have been included to provide an emphasis to prospective users and groups who may lack the computational infrastructure, while excluding tools that may solely require the deployment to local servers and experienced computational knowledge. Furthermore, tools reliant on data collected for more than 10 years were excluded. Additionally, implementations with a focus on humans were selected.

## 2. MicroRNA Biogenesis and Targeting

MiRNAs are small non-coding single-stranded RNAs encoded within non-coding sequences of the genome; however, a minority of miRNAs are known to also be encoded within exons [3]. Interestingly, miRNAs located within intronic regions of protein-coding genes can be transcribed from the same promoter as a host gene, while it has also been acknowledged that miRNAs may possess more than one transcription start site (TSS) [4,5].

The majority of miRNAs are transcribed by RNA polymerase II (Pol II), with RNA Pol II-related transcriptional regulators required for mediating this process [6]; however, in the case of some viral miRNAs, transcription can occur through RNA polymerase III (Pol III) [7]. The result of this process is a long stem–loop structure that is almost 1 kb in length, known as a primary miRNA (pri-miRNA). During the canonical mechanism of miRNA biogenesis, the pri-miRNA cleavage is processed through the microprocessor complex, which consists of the protein of the RNase III enzyme Drosha and DGCR8 (Pasha), resulting in the cleavage of the stem–loop construct to form a smaller precursor miRNA (pre-miRNA) molecule of around 65 nucleotides, with a hairpin shape [8].

Notably, miRNAs can be formed through alternative pathways to canonical miRNA biogenesis which are regarded as non-canonical [9]. Most non-canonical miRNAs are mirtrons, which are formed from an entire intron of the host gene transcript [10,11]. Here, during the non-canonical pathway, biogenesis is driven by Drosha/DGCR8-independent mechanisms, where the pre-miRNA is formed through the splicing of an entire intron of a host gene transcript [12] (Figure 1).

The newly formed pre-miRNA is then exported to the cytoplasm through the transport complex formed by the protein exportin5 and GTP-binding nuclear cofactor Ran-GTP. After translocation, the pre-miRNA-protein disassembly is induced through the hydrolysis of GTP present in Ran-GTP, resulting in the release of the pre-miRNA into the cytoplasm [13,14]. The pre-miRNA is then processed by the RNase III enzyme Dicer, leading to the production of a smaller double-stranded miRNA molecule [15]. Through the interaction between the Dicer and TAR RNA-binding protein (TRBP), further cleavage will occur, resulting in an miRNA duplex structure [16]. The final processed double-stranded miRNA is bound by the Argonaute protein (AGO), leading to the generation of the RNA-induced silencing complex (RISC). The AGO-bound miRNA duplex will then unwind, with either the separated 5p or 3p strand selected and incorporated into the mature RISC complex. The AGO-independent miRNA strand is not always degraded, and the 5p or 3p strands of many miRNAs can be detected in different tissues and processed independently [17]. Importantly, with the full activation of the RISC complex, post-transcriptional gene silencing can be undertaken by the mature RISC-bound miRNA.

Target mRNA recognition is achieved through complimentary (or semi-complimentary) matching between the miRNA seed sequence (a 2–7 nucleotide length sequence from the 5′ end of an miRNA) and the target site primarily located within the 3′UTR (untranslated region) of the target mRNA. Post-transcriptional gene silencing by miRNA regulation can occur through either the degradation of its target mRNA or through the process of translation repression. It has been reported that mRNA degradation accounts for 66–90% of miRNA-modulated gene silencing [18]. Here, the target mRNA is first deadenylated by the protein complexes PAN2-PAN3 and CCR4-NOT deadenylase [19]. Following the deadenylation of the mRNA transcript, this will promote the decapping of the mRNA by DCP1/DCP2, allowing for the final mRNA degradation step by the major cytoplasmic nuclease XRN1 [20,21]. While the full mechanism behind translational repression is less clear compared to that of mRNA degradation, it has been revealed through ribosome profiling that the inhibition of cap-dependent translation at initiation through RNA helicases such as eukaryotic translation initiation factors 4A (eIF4A) is involved [18] (Figure 1). 

Recent studies have demonstrated that some miRNAs are not actively engaged in target repression and instead have regulatory functions beyond the classic targeting dogma [22,23]. Specifically, these pools of miRNAs can regulate cell processes through direct miRNA–protein interaction [24]. Additionally, interactions between miRNAs and other classes of RNA apart from mRNAs haven been observed [25]. MiRNA expression itself is regarded as tissue-specific, therefore posing further considerations for scientists wishing to study particular miRNAs of interest [26]. Therefore, altogether, this demonstrates the complexities of miRNA functionality, which often pose further difficulties for developing accurate bioinformatic tools to aid researchers.

## 3. Tools for miRNA Sequences and Annotations

A significant asset to the process of miRNA identification has been the access to high-throughput technologies, resulting in the exponential detection and profiling of novel miRNAs at unprecedented levels [27]. Consequently, thousands of miRNAs have been identified across a vast array of species, which has dramatically empowered the field of miRNA research [28,29,30,31]. While advantageous, the need to store data regarding all novel and known miRNAs is equally important to allow researchers to access and view information ranging from sequence data to the genomic location and species annotation (Figure 2). Additionally, with the field continuously expanding, such data must be managed and updated for researchers interested in the latest information. Examples of databases for categorizing miRNA-related data are discussed here (Table 1). 

miRBase is regarded as one of the central repositories for the management and cataloging of miRNA data, which include sequence and annotation data supported by experimental evidence [32,33,34]. miRBase allows for searching for published pre-miRNA hairpin and mature miRNA sequences, in addition to readily available annotation and sequence data for downloading, to aid researchers. Currently, in the latest release (Release 22.1), there are 38,589 entries across 271 species representing hairpin precursor miRNAs, resulting in the production of 48,860 different mature miRNAs [35]. Compared to the last release (Release 21), this represents over one-third more sequences [35]. With the miRBase Registry service, the assignment of names for novel miRNA sequences is made before publication. Currently, for the human genome, there are 1900 entries consisting of annotated hairpin precursors, resulting in 2600 mature miRNA sequences.

Rfam comprises a library of various RNA families including non-coding RNAs, where each RNA family includes multiple sequence alignments and implemented covariance models [36,37]. RNA family-related information can be derived from either the literature or the user community. Rfam 14.9 represents the latest update to the database, containing information on around 4000 RNA families, which incorporate knowledge on seed alignments, structural information, species and additional database references [37].

RNAcentral represents an integrated database dedicated to non-coding RNA sequences by acting as a single platform providing access to over 18 million non-coding sequences from a diverse range of organisms [38]. Through RNAcentral, users may search for sequence-related information and annotation-related data such as precise locations within the genome, RNA modifications, miRNA–target interactions and ortholog and paralog information. RNAcentral aims to provide researchers with open access to high-quality non-coding RNA sequence data to enable interested audiences to access tools and resources regarding known individual sequences [39].

## 4. Tools for the Identification of Validated/Predicted miRNA Targets

It is well established that miRNAs can have roles extending across a wide range of cellular processes, and some have a significant impact in the context of disease. With a single miRNA being able to target potentially thousands of mRNAs and, in turn, one mRNA transcript possibly being regulated by several different miRNAs, this demonstrates the difficult task in understanding the complex nature of miRNA–target interactions (MTIs) occurring within the cell. Despite vast progress being made in recent decades, a large number of existing miRNA targets have yet to be discovered. Notably, the process of validating all prospective miRNA targets experimentally is both laborious and expensive; therefore, alternative means of investigating MTIs, such as bioinformatic-based approaches, have received increased interest and pursuit [72].

Various bioinformatic tools and algorithms have been developed to predict effective interactions between an miRNA and its potential target to address this issue. Examples of standard features taken into consideration for the development of prediction tools include free energy representing the measure of stability between the binding of miRNA and the candidate target mRNA, target site accessibility for the miRNA to bind to the target mRNA, evolutionary sequence conservation of miRNA target sites among species and miRNA seed sequence compatibility with the target mRNA [73,74]. While the use of such prediction tools supports the identification of candidate MTIs to be pursued for experimental validation [72], often, these tools are used to identify a list of MTI that are then included in a functional analysis to predict the biological role of an miRNA.

miRTarBase was first established in 2010 and represents one of the largest databases dedicated to collecting validated MTIs, which is continuously updated through surveying articles [40,41,42,43,44]. miRTarBase aims to provide users with a more comprehensive collection of experimentally supported MTIs, along with the integration with biological data from other existing databases. Collected MTIs are validated experimentally using reporter assays, western blots, microarray-based experiments involving the overexpression or knockdown of the miRNA and high-throughput sequencing. While many databases exist for collating miRNA data and MTIs, miRTarBase implements both manual and computational-based curation methods to provide researchers with accurate coverage of experimental evidence alongside additional data capturing the MTI. The first release (Release 1.0) comprised 657 miRNAs and 2297 miRNA target genes from a collection of 985 articles [40]. Currently, this has expanded to 19,912,394 experimentally validated MTIs occurring between 4630 miRNAs, and 27,172 miRNA target genes have been manually curated from a total of 13,389 articles and CLIP-seq data. In addition to updating data from pre-integrated databases such as miRBase [35] and HMDD [54], other databases are integrated into miRTarBase with each update. Data taken from databases such as EVmiRNA [75], miREDiBase [76], dbSNP [77], GWAS Catalog [78], ClinVar [79] and COSMIC [80] were integrated together recently in miRTarBase to implement and improve information regarding miRNA modifications [45]. Other important updates included in the most recent release relate to the enhancement of the text mining system through the optimization to provide a more accurate and sensitive approach for the increased recognition of MTIs and for facilitating later manual stages of collating information [45].

TargetScan was initially launched in 2003, allowing for the prediction of miRNA biological gene targets by searching for the presence of conserved 6 mer, 7 mer and 8 mer sites that match the seed region of each miRNA [46,47]. TargetScan allows users to search according to both miRNA and gene names across multiple species, including humans. Additionally, search predictions can be performed with poorly and broadly conserved sites. Predicted gene targets can be ranked by a cumulative weighted context++ score or Aggregate P_CT_ score, representing the expected efficacy of the sites and preferential site conservation, respectively [46]. Sorting by aggregate P_CT_ can provide insight into likely effective miRNA–target interactions, which, probably, in turn, may be more biologically crucial for the organism of interest [48]. The context++ score ranking can be of use for predictions across all interactions covered by TargetScan, such as miRNAs that are not highly conserved [46,49]. Another consideration for researchers utilizing TargetScan predictions is whether there is evidence of both miRNA and predicted target mRNA being expressed in the particular cells of interest [46].

DIANA-TarBase was first launched in 2006 as one of the first available databases containing experimentally validated miRNA–mRNA interactions [50]. At present, there are over a million entries covering 665,843 miRNA–mRNA interactions from more than 33 low-yield and high-throughput experiments, of which 516 different cell types and 85 tissues are included within the 18 species on the database [51]. DIANA-TarBase v8.0 allows users to upload a query with miRNA or a list of gene names to identify interactions with the addition of filtering for species, tissues/cell types, the experimental methodologies utilized, the publication year or the in silico predicted score [51].

miRDB is an online database for primarily miRNA prediction and functional annotations, providing searches for a total of five species, including humans and mice [52]. Here, all of the miRNA targets are predicted using miRTarget [53], the latest prediction algorithm, which was generated by analyzing miRNA and gene target interactions across various RNA-seq and CLIP-seq datasets. miRDB also allows users to flexibly search for miRNA targets through the inclusion of submitting users’ own sequences, both miRNA and gene targets. Equally, miRDB provides the option of performing searches within the coding region or 5-UTR in the case of the mRNA gene target. Additional features implemented include the ability to predict cell-specific miRNA targets using the target expression profiles of approximately 1178 cell lines from RNA-seq analysis from Expression Atlas [81]. The latest release of miRDB comprises 2656 human miRNAs predicted to regulate 1,610,510 gene targets [52].

## 5. Tools Related to Human Diseases

As miRNAs are involved in biological functions and signaling pathways, it has been observed that the dysregulation of miRNAs is associated with various human diseases, including cancer [82,83,84,85]. Importantly, the understanding of MTIs in the context of disease will aid biomedical researchers working towards solutions for the diagnosis, treatment or prevention of certain disorders, therefore highlighting the clinical applications miRNAs may have for patients [86]. Currently, there are databases that solely exist to accumulate and collate information into a single platform regarding miRNA involvement in several diseases (Table 1).

The Human MicroRNA Disease Database (HMDD) is a repository dedicated to experimental-based evidence covering human microRNA–disease associations [54,55,56]. HMDD provides users with easy access to search, browse and analyze experimental data supporting miRNA associations in a range of diseases. Additionally, researchers can contribute to the community through the submission feature on the web interface, allowing for submitting novel miRNA data onto the database. Unique features made available on the database are parameters such as disease spectrum width (DSW) score and miRNA spectrum width (MSW). DSW was devised to evaluate an miRNA’s effect in human diseases, allowing for an understanding of the importance of one miRNA across human diseases [57]. The MSW parameter gives a score based on the number of miRNAs associated with a particular disease, providing insight into the complexity of that human disease [54,56]. HMDD version 3.2 has accumulated 35,547 miRNA-disease association-based entries comprising 1206 miRNA genes and 893 diseases from 19,280 papers [56].

OncomiR allows for the exploration of miRNA dysregulation in the context of cancer [58]. Users can search according to an miRNA of interest to gain insights into known associations, with processes ranging from tumor formation to patient survival outcome and particular gene targets specific to the cancer of interest. Similarly, researchers may search via a particular cancer of interest to understand the miRNA expression profile from their selection. One of the features provided by OncomiR is the ability to select clinical parameters such as sex, tumor stage and grade to observe whether particular stages of cancer result in the variation in the associated miRNAs. OncomiR contains miRNA-seq and RNA-seq data comprising approximately 1200 mature miRNAs and 30,000 miRNA target genes from The Cancer Genome Atlas (TCGA) [87].

The database of Differentially Expressed MiRNAs in human Cancers (dbDEMC) represents an integrated database that is freely available for researchers interested in differentially expressed miRNAs in various cancers [59]. dbDEMC has been extended to cover additional species, including mice and rats, in addition to human data, benefiting researchers wishing to apply such analysis within these model organisms. dbDEMC consists of 3268 miRNAs, of which 40 cancer types are included from 403 datasets [60]. From the 3268 miRNAs included in dbDEMC, 2584 differentially expressed miRNAs are human, representing the majority of publicly available miRNA expression datasets from resources such as Gene Expression Omnibus (GEO) [88], ArrayExpress [89] and additional miRNA-seq based data from the Sequence Read Archive (SRA) [90]. The Cancer Genome Atlas (TCGA) [87] is also incorporated into the database to improve the quality of the miRNA and gene target information captured.

## 6. Tools for Pathway Identification

As discussed previously, miRNAs are associated with various biological processes and signaling pathways. The interpretation of high-throughput datasets relies heavily on functional enrichment tools that incorporate high-quality annotations of genes, proteins, chemicals and pathways. The Gene Ontology (GO) [91,92] provides one of the most commonly used resources included in these enrichment tools. It is very often used to interpret non-coding RNA (ncRNA) transcriptomic data, including miRNAs [93,94]. The biological domains described by GO enable the normal activity of gene products, their role in cellular pathways and processes and the location of these products in the cellular environment to be captured (for example, miRNA-mediated gene silencing). However, the interpretation of such datasets using functional enrichment tools is often limited by the availability of the annotation data from the literature that can be included in the analysis. Researchers often turn to tools that predict the targets of the RNAs and then conduct the enrichment analysis on the predicted targets rather than the ncRNAs [93]. Recognizing the potential misleading interpretations that can arise from this approach, the GO Consortium and some molecular interaction databases include capturing the functional role of miRNAs, experimentally verified interactions between miRNAs and their targets and the impact of these interactions on downstream cellular and organism-wide processes [38,93,95,96]. To date, there are over 27,500 human ncRNAs annotated in the GO annotation files, providing around 65,000 GO annotations. Of these, 200 miRNAs have been associated with at least one target mRNA; consequently, 800 direct MTIs have been captured.

Functional analysis is also achieved through the Kyoto Encyclopedia of Genes and Genomes (KEGG) database. KEGG represents eighteen integrated databases aiming to enable the computational reconstruction of biological systems, with content ranging from genes to proteins, chemical substances and reactions, drugs, molecular interactions and human disease [97]. The PATHWAY database is the central database within KEGG [97,98], consisting of manually drawn reference pathway maps integrated with known organism-specific pathways that are computationally assigned. Currently, a total of 8630 organisms are captured through KEGG, including humans, in over 562 reference pathway maps. Recently, multiple databases have been established to capture the regulation of signaling pathways by miRNAs and their involvement in human disease due to the promise of therapeutic intervention. Among these present databases, a few are discussed in this review (Table 1).

DIANA-miRPath allows for the real-time analysis of KEGG molecular pathways or GO terms across seven species [61]. Integrated into the database are additional interactions from other resources, such as Targetscan [46] and DIANA-TarBase [51]. DIANA-miRPath also provides the option of performing The Fisher’s Exact Test, EASE score and False Discovery Rate with unbiased empirical distributions [61]. In addition, DIANA-miRPath utilizes gene and miRNA annotations from platforms such as Ensembl [99] and miRBase [35], with information regarding single-nucleotide polymorphism (SNP) locations from the database dbSNP [77]. One feature of DIANA-miRPath is the reverse-search module, allowing for identifying miRNAs regulating specific biological pathways based on GO term enrichment analysis. Here, details regarding the miRNA, enrichment *p*-values and genes targeted, along with links to further information, are provided on the interface.

The miRNA Pathway Dictionary Database (miRPathDB) provides users with freely available access regarding biological pathways and functional categories from GO terms associated with certain miRNAs supported through predicted and experimental evidence [62,63]. It includes general information governing the miRNA, such as the chromosomal positions, stem–loop precursor and mature miRNA sequences, along with known targets and significant pathway information. miRPathDB additionally provides third-party links for each miRNA to external database sources such as miRBase [35], miRCarta [100], TargetScan [46] and TissueAtlas [101]. The latest release (Release 2.0) displays information on 27,452 human miRNAs, 28,352 target mRNA genes and a total of 16,833 pathways [63], representing more than a ten-fold increase in information through the integration of the databases miRBase [35] and miRCarta [100].

miTALOS is an interactive web resource providing researchers with insights into signaling pathways regulated by miRNAs, with the addition of integrated tissue filters [64,65]. This is especially useful for those interested in miRNA regulation, as it is well-acknowledged that miRNAs exhibit differing tissue expression profiles. Taking tissue or cell expression into consideration during functional analysis may reduce the likelihood of inaccurate miRNA-pathway enrichments. Therefore, through filtration, one can browse the biological effect of an miRNA of interest in the context of an available cell line or tissue. In addition, users can perform an analysis on both human and mouse miRNAs. Currently, miTALOS v2 provides coverage of 1562 miRNAs, which span 42 different tissues in humans [65].

## 7. Tools for SNP Effect Prediction

MiRNA targeting is heavily reliant on the complementarity between the target mRNA binding site and the miRNA seed sequence. Single-nucleotide polymorphisms (SNPs) are among the most common sources of genetic variation in the genome, with most reported ones occurring within non-coding regions [102]. A variation within an miRNA sequence may impact the functionality of the miRNA, if it affects the stability of the MTI. An SNP that lies in a regulatory region and influences the expression level of an miRNA may also affect the expression levels of the miRNA targets [103,104,105]. Importantly, such profound effects may, in turn, contribute to disease, therefore leading to an increased focus in recent years on genetic variants within miRNAs and miRNA–target binding sites. This is reflected through the existence of tools aimed at investigating such effects, a few of which are described here (Table 1).

Polymorphism in the microRNA Target Site (PolymiRTS) database represents a collection of DNA polymorphisms within miRNAs and miRNA target sites across mouse and human genomes, allowing for the identification of possible transcriptional and phenotypic variation [66,67,68]. PolymiRTS, at present, focuses on polymorphisms present in miRNA seed regions and target sites, with less emphasis on other aspects such as pre-miRNAs, pri-miRNAs and miRNA regulatory regions, which have also been reported to be associated with various diseases [68]. PolymiRTS version 3.0 has incorporated the KEGG database, allowing users to observe biological pathways for humans and mice to establish associations between polymorphisms located in miRNA target sites and functional effects within various biological processes. Additional features that may be of use to users include the incorporation of CLASH experimental datasets [106], allowing for the extended capturing of the number of miRNAs and target sites; the PolymiRTS database includes 18,514 high-confidence canonical and noncanonical targets of 399 human miRNAs [68].

miRNASNP was first developed in 2011 to provide a platform dedicated to the characterization of SNPs that may impact processes related to miRNA functionality, such as miRNA biogenesis or miRNA target binding [69,70]. The current release, miRNASNP-v3 [71], has integrated human single-nucleotide variation data from a range of additional databases such as dbSNP [77], GWAS Catalog [78], COSMIC [80] and ClinVar [79], allowing for the expansion of the number of SNPs captured by miRNASNP. miRNASNP-v3 has captured information on 7,161,741 SNPs and 505,417 DRVs (disease-related variations) on 2630 mature miRNAs and 18,152 target gene 3′UTRs [71]. Examples of features available on miRNASNP include allowing users to explore predicted and experimentally verified effects of SNPs on MTIs, observe reported associations between miRNA-related SNPs and human disease and browse the impact of SNPs on the pre-miRNA secondary structure [71].

## 8. Conclusions

The first miRNA, *lin-4,* was discovered in 1993 in *Caenorhabditis elegans*, leading to one of many revolutions in molecular biology [107,108]. Since then, our understanding of the importance of miRNAs has grown exponentially, with numerous miRNAs being identified across various organisms. Similar to miRNAs, other ncRNAs have been acknowledged to also play roles within disease regulation, highlighting the importance of ncRNAs overall. Along with the vast progress in identifying novel miRNAs, bioinformatic tools and repositories have been established to capture and manage the latest miRNA-related data.

In this review, numerous available bioinformatic resources have been introduced, which have been developed to aid researchers interested in the complex nature of miRNA regulation. However, each tool will have its own individual advantages and disadvantages, which is important to consider. One such limitation that is well acknowledged in web-based tools and databases is the accumulation of false-positives due to the increased sensitivity. A proposed solution for overcoming this problem is through the implementation of machine learning and using filter-based algorithms to reduce the rate of false outputs [109,110]. This highlights the importance of future tools which can integrate multiple computational algorithms to deliver optimal results for miRNA researchers. As further data are generated through high-throughput technologies, the reliance on such tools will further increase in the future, with user-friendliness likely to be a central factor in their successful use for analysis and interpretation. Finally, with miRNAs being recognized as potentially promising targets for therapeutic intervention, such bioinformatic resources will also play roles in future endeavors within this type of research.

## Figures and Tables

**Figure 1 ncrna-09-00018-f001:**
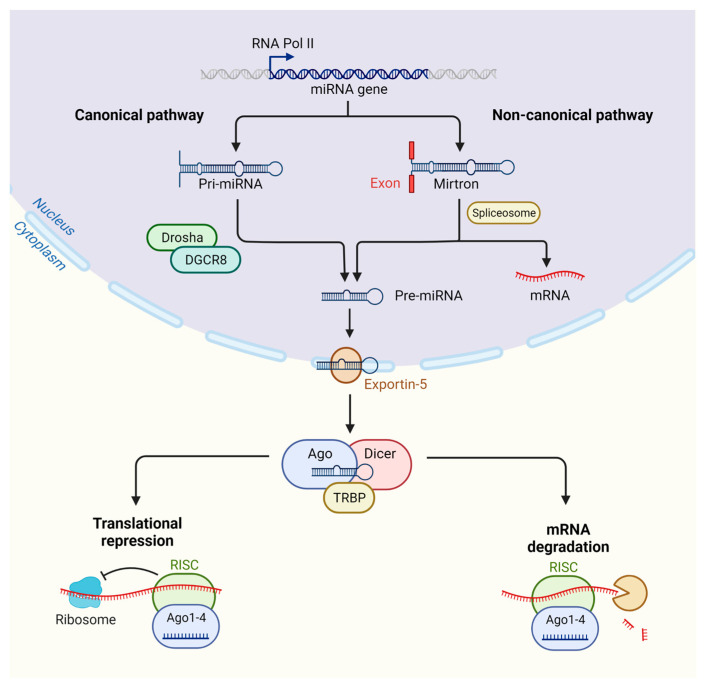
Overview of the biogenesis of miRNAs in animals. After RNA Pol II has transcribed the miRNA gene, an miRNA may undergo further processing through the canonical or non-canonical mechanism of biogenesis. In the canonical pathway, a pri-miRNA will be processed through the microprocessor complex, composed of Drosha and DGCR8 to form a pre-miRNA. The non-canonical pathway, on the other hand, is regarded as Drosha/DGCR8-independent with a pre-miRNA formed through splicing from the entre intron. After this stage, the pre-miRNA may enter to join the canonical pathway. Upon being translocated to the cytoplasm through exportin-5, along with GTP-binding nuclear cofactor Ran-GTP, pre-miRNA is then processed by the RNase III enzyme Dicer, leading to the production of a smaller double-stranded miRNA molecule. Through the interaction between the Dicer and TAR RNA-binding protein (TRBP), further cleavage will occur, resulting in an miRNA duplex structure. The final processed double-stranded miRNA is bound by the Argonaute protein (AGO), leading to the generation of the RNA-induced silencing complex (RISC). The AGO-bound miRNA duplex will then unwind, with either the separated 5p or 3p strand selected and incorporated into the mature RISC complex. (Created with BioRender.com).

**Figure 2 ncrna-09-00018-f002:**
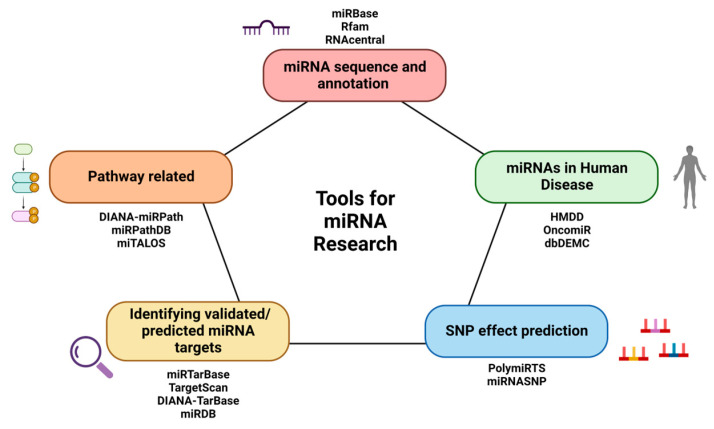
Schematic of available online microRNA-based bioinformatic tools. (Created with BioRender.com).

**Table 1 ncrna-09-00018-t001:** Overview of described tools available for miRNA analysis.

Category	Tool Name	Organism	Last Update	Summary Features	URL	References
miRNA sequences and annotation	miRBase	271 species, including humans	2019	Central primary repository for the storage of miRNA annotation and sequence data Nomenclature may differ according to the version of miRBase utilized, especially in the case of tools incorporating data from miRBase	https://www.mirbase.org/	[32,33,34,35]
	Rfam	9 species, including humans	2022	Allows users to submit a novel RNA family to ensure up-to-date annotations Synchronization of the majority of miRNA families across miRBase, Rfam and RNAcentral is currently in progress.	https://rfam.xfam.org/	[36,37]
	RNAcentral	>800,00 species, including humans	2022	A public resource capturing information on all non-coding RNAs, including miRNAs Comprehensive and actively updated	https://rnacentral.org/	[38,39]
Target Discovery	miRTarBase	37 species, including humans	2021	One of the largest databases of experimentally validated MTIs Regulators of miRNAs are included in the database	https://mirtarbase.cuhk.edu.cn/	[40,41,42,43,44,45]
	TargetScan	13 species, including humans	2018	Can distinguish between different isoforms of a particular gene target during scoring TargetScan conducts its own classification of miRNA families, based on the identical seed region, so the nomenclature can differ between databases such as miRBase	https://www.targetscan.org/vert_72/	[46,47,48,49]
	DIANA-TarBase	18 species, including humans	2017	Places focus on experimentally validated MTIs across 18 species Over 1 million entries, consisting of 670,000 unique, experimentally supported MTIs Tedious interface layout when investigating reference links and MTIs	https://dianalab.e-ce.uth.gr/html/diana/web/index.php?r=tarbasev8	[50,51]
	miRDB	5 species, including humans	2019	Allows for custom prediction through the submission of user-provided miRNA or gene target sequences	http://mirdb.org/	[52,53]
Human disease-related	HMDD	Humans	2019	User-friendly interface offering clear explanatory notes and references for users HMDD is only limited to collating data surrounding humans only and ignores model organisms such mice and rats	http://www.cuilab.cn/hmdd	[54,55,56,57]
	OncomiR	Humans	2017	Clear walkthrough page with guidance for users, with all features clearly defined Users are able to explore for data capturing almost 10,000 patients across 30 cancer types	http://www.oncomir.org/	[58]
	dbDEMC	3 species, including humans	2021	Offers analysis on additional species other than humans (mice and rats) Currently, 40 cancer types and 149 cancer subtypes are captured	https://www.biosino.org/dbDEMC/index	[59,60]
Pathway-related	DIANA-miRPath	7 species, including humans	2015	Unique reverse search module allowing for the identification of miRNAs enriched within a particular pathway Offers analysis extended to a further list of species (humans, mice, rats, dogs, flies, worms and chickens) Of the remaining tools described under this category, DIANA-miRPath represents the oldest updated tool	https://dianalab.e-ce.uth.gr/html/mirpathv3/index.php?r=mirpath	[61]
	miRPathDB	2 species, including humans	2018	Clear user-friendly layout, with clear explanatory notes and an available guide Places focus on two species: humans and mice	https://mpd.bioinf.uni-sb.de/	[62,63]
	miTALOS	2 species, including humans	2016	First resource to offer a tissue-specific filter Three major pathways are integrated: KEGG, Wiki Pathways and Reactome	http://mips.helmholtz-muenchen.de/mitalos/#/search	[64,65]
SNP effect prediction	PolymiRTS	2 species, including humans	2014	Provides an advantage through the analysis of mouse data Last known update of PolymiRTS was in 2014.	https://compbio.uthsc.edu/miRSNP//	[66,67,68]
	miRNASNP	Humans	2020	Compared to PolymiRTS, it contains fewer available features on the web interface for users MiRNASNP is kept continuously up to date	http://bioinfo.life.hust.edu.cn/miRNASNP/#!/	[69,70,71]

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
