# Peer review of "Insights into Online microRNA Bioinformatics Tools"

_ncrna, 2023, doi:10.3390/ncrna9020018_

Round 1

Reviewer 1 Report

Manuscript by Buitrago et al. covers important topic in RNA research: a description and comparison of available bioinformatics tools for analysis of miRNAs. The overview of available tools is comprehensive and valuable.

My main comment is that the complexity of miRNA-mediated mechanisms should be mentioned in more detail in the section 2. I find this important as justification of difficulties in developing of accurate bioinformatic tools. For example it could be mentioned that miRNAs can also target ORF regions; they can bind to other RNAs, apart from mRNAs; multiple binding sites are important for miRNA-based gene silencing; miRNA expression is often tissue-specific, etc.

Minor comments:

I suggest re-check the manuscript for clarity of the statements, for example:

-          Abstract, line 10: word “increased” is repeated in the sentence and I find “in recent years” misleading here. The field is constantly growing since two decades.  

-          Introduction, line 20: the part of the sentence: “(…) which have allowed…” is redundant.

Other minor suggestions:

-          ~19 references (form 106) are from the last 3 years – please reconsider if all relevant novel papers are included (of course fundamental older publications should be also included).

-          Lines 42-44: Microprocessor complex could be mentioned.

-          Line 50: snoRNA are not miRNAs, there are some examples of snoRNA-derived miRNA

Reviewer 2 Report

Review report on “Insights into online microRNA bioinformatics tools”

This review article provides a brief overview of the current web-based resources available for miRNA research, including miRNA sequences, target prediction/validation, miRNAs associated with disease, pathway analysis, and genetic variants within miRNAs. The paper is clearly written. However, there is room for improvement concerning the documentation style of the paper. I list some concerns as follows:

1.     According to [2], there are more than 1000 miRNA bioinformatics tools, many of which allow web-based applications. But the authors do not clearly state their criteria for selecting the web-based tools presented in the review.  Please provide the criteria and  justify them.

2.     While the authors compiled a list of online tools for different purposes, they do not provide a comprehensive comparison of their advantages and disadvantages, thus making it difficult for readers to choose the most suitable tool for their research needs. To address these concerns, the authors could consider providing more guidance on tool selection and including a comparison of the different tools in Table 1.

All in all, I recommend the paper to be major revision. I include more precise suggestions for changes below.

Use either "MiRNAs" or "microRNAs" consistently throughout the text.

Page 1, Line 11. “Due to such growth in the area, developing multiple new databases addresses various biological questions regarding miRNA research.”  the sentence reads odd and needs to be rephrased for clarity.

Page 1, line 19, "derive" should be "derives".

Page 2, consider including a schematic diagram to explain the microRNA biogenesis process, which would greatly enhance the reader's understanding.

Page 2, line 57: "leading to the production" should be "leading to the production of"

Reviewer 3 Report

MicroRNAs are known to regulate gene expressions at the post-transcriptional level and are functionally important in diverse biological and pathological processes. In this manuscript, the authors summarized current available computational tools and databases for researchers better utilize the resources to make impact findings towards understanding the functional roles of MicroRNAs in biology. The manuscript is well written and the summary is comprehensive. 

Round 2

Reviewer 2 Report

In my opinion, most of my previous concerns are addressed satisfactorily and/or clarified properly.

I find the revisions adequate and have no further comments.